# Electrospun Poly(lactic acid) and Silk Fibroin Based Nanofibrous Scaffold for Meniscus Tissue Engineering

**DOI:** 10.3390/polym14122435

**Published:** 2022-06-16

**Authors:** Siripanyo Promnil, Chaiwat Ruksakulpiwat, Piya-on Numpaisal, Yupaporn Ruksakulpiwat

**Affiliations:** 1School of Polymer Engineering, Institute of Engineering, Suranaree University of Technology, Nakhon Ratchasima 30000, Thailand; siripanyo.pn@gmail.com (S.P.); charuk@sut.ac.th (C.R.); 2Center of Excellence on Petrochemical and Materials Technology, Chulalongkorn University, Bangkok 10330, Thailand; 3Research Center for Biocomposite Materials for Medical Industry and Agricultural and Food Industry, Suranaree University of Technology, Nakhon Ratchasima 30000, Thailand; 4School of Orthopaedics, Institute of Medicine, Suranaree University of Technology, Nakhon Ratchasima 30000, Thailand

**Keywords:** PLA, silk fibroin, electrospinning, tissue engineering, gene expression, cell viability

## Abstract

Biopolymer based scaffolds are commonly considered as suitable materials for medical application. Poly(lactic acid) (PLA) is one of the most popular polymers that has been used as a bioscaffold, but it has poor cell adhesion and slowly degrades in an in vitro environment. In this study, silk fibroin (SF) was selected to improve cell adhesion and degradability of electrospun PLA. In order to fabricate a PLA/SF scaffold that offered both biological and mechanical properties, related parameters such as solution viscosity and SF content were studied. By varying the concentration and molecular weight of PLA, the solution viscosity significantly changed. The effect of solution viscosity on the fiber forming ability and fiber morphology was elucidated. In addition, commercial (l-lactide, d-lactide PLA) and medical grade PLA (pure PLLA) were both investigated. Mechanical properties, thermal properties, biodegradability, wettability, cell viability, and gene expression of electrospun PLA and PLA/SF based nanofibrous scaffolds were examined. The results demonstrated that medical grade PLA electrospun scaffolds offered superior mechanical property, degradability, and cellular induction for meniscus tissue regeneration. However, for commercial non-medical grade PLA used in this study, it was not recommended to be used for medical application because of its toxicity. With the addition of SF in PLA based scaffolds, the in vitro degradability and hydrophilicity were improved. PLAmed50:SF50 scaffold has the potential to be used as biomimetic meniscus scaffold for scaffold augmented suture based on mechanical properties, cell viability, gene expression, surface wettability, and in vitro degradation.

## 1. Introduction

Poly(lactic acid) (PLA) is a linear aliphatic polyester that was originally obtained from natural crops [1]. Lactide is an intermediate for the production of the high molar mass PLA via ring opening polymerization (ROP). This monomer has large importance because it controls the synthesis of polymer production. The monomer exists as two stereo isomers, l-lactide and d-Lactide [2]. The processing, crystallization, and degradation behavior of PLA all depend on the stereochemical structure and composition of the polymer chains, which is influenced by the lactide isomers [3]. The presence of l- or d-lactide monomer affects the physical and mechanical characteristics of the final polymer. PLLA polymer has the highest melting point among the other PLA forms due to its crystalline nature. As D-isomer is incorporated in the PLA chain, it reduces its crystallinity, lowering the melting points of PLA copolymers. In addition to that, PLLA (higher crystallinity) is often more used than PDLA [4]. PLA and its copolymers were developed as biomedical materials based on their bioabsorbable and biocompatible properties and have been widely used in orthopedic surgery including sutures, prostheses, and tissue engineering scaffold in which high- and low-molecular-weight PLAs are utilized [5].

Due to the manufacturing process, the cost of medical grade PLA is much higher than the commercial grade PLA. It must be synthesized under a physician’s license to pass the validation requirements of the regulatory agencies. This meant that polymer manufacturers would need to have controls over their design and development processes, including strict controls of the raw materials and components used to manufacture the finished product [6]. Thus, it is interested to compare the scaffold properties obtained from medical grade PLA (pure PLLA) to the commercial grade PLA (l-lactide, d-lactide PLA) at the comparable molecular weight in order to study the difference of physical and mechanical properties of PLA and PLAmed scaffolds. However, PLA has slow degradation, poor hydrophilicity, and poor cell adhesion. To eliminate the drawback of PLA, a combination of PLA and silk fibroin (SF) scaffold was studied [7].

Silk is a fibrous protein produced by silkworms. Silk consists of two components; the main part is fibroin, which is the core fiber and the coverage protein called sericin [8,9]. Silk fibroin is a biocompatible natural polymer which is non-toxic, immunogenically inert and provides good elasticity. Moreover, it has been fabricated and used as the medical suture and has a long safety record. Therefore, SF is a good candidate natural polymer for our PLA composite scaffold in meniscus tissue engineering [10]. PLA/SF composite scaffolds have good cell compatibility and are conducive to cell adhesion and growth [11,12,13,14]. The presence of silk fibroin also significantly enhanced the enzymatic degradation ability of the PLA matrix, which is good for bone tissue engineering application [7,15].

Meniscus is a fibrocartilageous structure which covers the tibial articular surface. Meniscus functions to distribute, absorb, and transmit load across the knee joint. Meniscus also enhances knee stability by increasing joint congruity and contact surface area [16]. From this reason, meniscus is vulnerable to injury which caused a meniscus tear to become a common injury in the knee joint. Unfortunately, the efficacy of meniscus repair depends on location with respect to the vascular supply. Outer and middle meniscus is avascular and can heal with some fibroblasts. The inner avascular zone presents a limited amount of chondrocyte like cells that possess poor healing potential [17]. Since inner menisci have pool healing potential, an unhealed meniscus ends up with meniscus resection that causes high contact stress on the articular surface and leads to osteoarthritis [18].

For the aforementioned reason, tissue engineering and cell-based therapy have been proposed as a biological augmentation for meniscus repair. There are three main components in tissue engineering: cells, scaffolds, and bio-active molecules. Cells and bioactive molecules function simultaneously to produce new tissue formation. Scaffold is a cell shelter and delivery system. Scaffolds is made from biocompatible material and should support target tissue regeneration as well as provide mechanical competent [19]. Currently, a three-dimensional biomimetic scaffold which imitates a host tissue environment has been studied. Collagen is a major component of the meniscus extracellular matrix. Collagen fibers mostly align longitudinally in a circumferential orientation; some fibers lie perpendicularly as a radial fiber. Electrospinning is an interesting technique to fabricate the fibrous scaffold. Electrospun fibers can be fabricated on a micro to nano-scale with a great surface area and high porosity that is similar to a natural extracellular matrix (ECM) in both architecture and mechanical properties [20].

Our research aimed to develop a PLA/SF composite biomimetic meniscus scaffold using an electrospinning technique. The application of our scaffold will be used for a scaffold augmented suture. The effect of molecular weight and concentration of PLA on viscosity and fiber morphology was observed. Moreover, the effect of SF contents on fiber morphology, thermal properties, wettability, degradability, mechanical properties, and cytotoxicity of the PLA/SF scaffold were examined. Comparison of these properties between commercial grade PLA (l-lactide, d-lactide PLA) and medical grade of PLA (PLLA) based scaffold was also made.

## 2. Materials and Methods

### 2.1. Materials

Two commercial non-medical grade PLA with different molecular weight, PLA3251D (low molecular weight with 99% l-lactide and 1% d-lactide; PLAL [21]) and PLA4043D (high molecular wight with 94% l-lactide and 6% d-lactide content; PLAH [22]) provided by NatureWorks LLC(Minnetonka, MN, USA) and a medical grade, Resomer L209S (pure PLLA; PLAmed) from Sigma-Aldrich (Burlington, MA, USA.) were used. PLAH and PLAmed have a molecular weight in a comparable range. Chloroform RPE was purchased from Carlo Erba Reagents (Milano, Italy). Formic acid was obtained from Merck (Darmstadt, Germany). Bombyx mori cocoons were provided by Queen Sirikit Sericulture Center, Nakhon Ratchasima, Thailand. Sodium carbonate (${\mathrm{Na}}_{2}{\mathrm{CO}}_{3}$, analytically pure) and calcium chloride (${\mathrm{CaCl}}_{2}$, analytically pure) were purchased from Carlo Erba (Milano, Italy).

### 2.2. Silk Fibroin Preparation

Cocoons were degummed in ${\mathrm{Na}}_{2}{\mathrm{CO}}_{3}$ solution at 98 ± 2 °C for 30 min, rinsed with distilled water and dried overnight. Degummed silk fibers were dissolved in ${\mathrm{CaCl}}_{2}$ solution by stirring at 98 ± 2 °C for 1 h. Then, the SF aqueous solution was filtered to remove undissolved component, and dialyzed against distilled water for 3 days. The SF solution was filtered and lyophilized to obtain the SF powder.

### 2.3. Electrospinning Solution Preparation

PLA was dissolved in chloroform at various concentrations as indicated in Table 1. SF solution was prepared by dissolving SF in formic acid (12% *w*/*v*). The solution of PLA commercial grade and SF were mixed in three different ratios of PLAH:SF (75:25, 50:50 and 25:75). The solution of PLA medical grade and SF was mixed at a 50:50 ratio. The symbols used for PLA/SF samples were shown in Table 2. The emulsion was created by magnetic stirring for 12 h to obtain uniform emulsion. The electrospinning parameters were as follows: a positive voltage of 20 kV, a collector distance of 15 cm, and the flow rate of 2.0 mL/h.

### 2.4. Characterization of PLA Solutions

The viscosity of PLA solutions was measured by a Brookfield rheometer (AMETEK Brookfield, Middleboro, MA, USA) with a cone-plate at constant temperature (25 °C).

### 2.5. Fiber Morphology

The microstructure of electrospun fibers was observed under field emission scanning electron microscopy (FESEM; Carl Zeiss Auriga, Oberkochen, Germany) with gold coating. The diameter of the fibers was measured from the micrographs using image analysis software (ImageJ) in 100 random fibers, and the diameter distribution histograms were plotted by OriginLab software.

### 2.6. Mechanical Properties

Tensile properties were determined by using an Instron Universal Testing Machine (Instron 5565, Norwood, MA, USA) with crosshead speed on 10 mm/min, 1 kN load cell at room temperature. The electrospun test specimens with 1 cm width and 10 cm original length were prepared. The reported data of tensile strength, elongation at break and Young’s modulus represent the average results from five test specimens (*n* = 5).

### 2.7. Thermal Properties

The thermal stabilities of PLA and PLA/SF scaffolds were analyzed by a TGA/DSC1 thermogravimetric analyzer (Mettler Toledo, Greifensee, Switzerland) at a heating rate of 10 °C/min under nitrogen from 25 to 500 °C. The thermal properties of scaffolds were analyzed by differential scanning calorimetry (DSC) on a Pyris Diamond DSC machine (PerkinElmer, Waltham, MA, USA) in nitrogen atmosphere. The samples were heated from 25 to 200 °C at a rate of 10 °C/min (first-heating scan). After keeping the specimens at 200 °C for 5 min, they were cooled to 25 °C at 10 °C/min. Then, they were heated again to 200 °C at 10 °C/min (second-heating scan). The glass transition temperature ($T_{g}$), the melting temperature ($T_{m}$), the cold crystallization temperature ($T_{\mathrm{cc}}$), the cold crystallization enthalpy ($\Delta H_{\mathrm{cc}}),$ and the melting enthalpy ($\Delta H_{m}$) were determined from the first and second heating scan. The melt crystallization temperature ($T_{c}$) and the crystallization enthalpy ($\Delta H_{c}$) were obtained from the cooling scan. The degree of crystallinity (%$\chi_{c}$) of PLA and biocomposites was determined by Equation (1):(1)$$\%\;\mathrm{Crystallinity}\;(\chi_{c})=[(\Delta H_{m})/(\Delta H_{m}^{0})]\;\times \;100\;\times \;1/W_{\mathrm{PLA}}$$
in which $\Delta H_{m}$ is the measured melting enthalpy (J/g) from the heating scan, $\Delta H_{m}^{0}$ is the theoretical melting enthalpy of completely crystalline PLA (93.7 J/g) [28,29], and W_PLA_ is the PLA weight fraction in the biocomposites.

### 2.8. Fourier-Transform Infrared Spectroscopy (FTIR)

The FT-IR spectra with characteristic absorption peaks of SF powder, PLA, and PLA/SF nanofibers were determined by a FTIR spectrophotometer (Bruker Tensor 27, Billerica, MA, USA). All samples were directly characterized in attenuated total reflectance (ATR-FTIR) mode in the spectral range of 4000–400 ${\mathrm{cm}}^{-1}.$

### 2.9. In Vitro Degradation

Scaffolds with dimensions of 1 cm width and 1 cm length (*n* = 3) were immersed in phosphate buffered saline (PBS, pH = 7.4) and incubated at 37 °C with 5% carbon dioxide (${\mathrm{CO}}_{2}$) for 14, 28, 42, and 84 days (2, 4, 6, and 12 weeks). PBS was changed every 3 days. The appearance of all scaffolds was observed every 2 weeks. The scaffolds were washed with distilled water, dried, and weighed. The percentage of residual weight was calculated by Equation (2) [30]:% Residual weight = 100 − [((W_i_ − W_f_)/W_i_) × 100](2)
where $W_{i}$ is initial weight of sample, while $W_{f}$ is the weight of sample after immersing in PBS.

### 2.10. Surface Wettability

Water contact angle was studied to assess the surface wettability properties of the PLA/SF electrospun nanofibers. Distilled water with controlled volume of 15 µL was dropped on the surface of each sample. After a 60 s exposure at ambient temperature, the images of water drop on the sample surface were recorded by a USB digital microscope (1600×) and analyzed with ImageJ software. Three different points (*n* = 3) were measured for each sample.

### 2.11. In Vitro Cell Culture Studies

Scaffolds were cut into 4 mm x 4 mm dimension (*n* = 3), separated into two groups. The first group was plunged into 70% alcohol, dried at 30 °C for 24 h, and sterilized under ultraviolet (UV) light for 30 min while the second group was not plunged into alcohol. The scaffolds were then placed in 96-well plates and 5 × 10^3^ human chondrocyte cells were seeded onto the scaffolds. The cell seeded scaffolds were cultured in Dulbecco’s modified Eagle’s medium (DMEM) with 10% fetal bovine serum, 1% l-glutamine, and 1% penicillin-streptomycin, put in 37 °C with a 5% ${\mathrm{CO}}_{2}$ humidified incubator. The cell viability was assessed with an MTT (3-[4,5-dimethylthiazol-2-yl]-2,5 diphenyl tetrazolium bromide) assay at days 1, 3, and 7. Optical density of each well was read at 590 nm using a microplate reader. The percentage of cell viability was calculated by comparing the absorbance of cells cultured on scaffolds to that of control Equation (3):% Cell viability = (O.D.of treatment)/(O.D.of control) × 100(3)

### 2.12. Quantitative Analysis for Gene Expression

According to cell viability results, PLAmed−6% and PLAmed50:SF50 scaffold were used to assess gene expression. The nanofibrous scaffold sheets were cut into a circle, 12 mm in diameter, and sterilized under UV light for 30 min. Then, the prepared scaffolds (*n* = 3) were put in 24 wells and incubated in culture media for 4 h. The HCPCs were seeded with 2.5 × 10^4^ cells in each scaffold. A cell seeded scaffold was then cultured for 7, 14, and 28 days. Total RNA was extracted from the HCPCs on the scaffolds for quantitative gene-analysis using RNeasy mini-Kit (Qiagen, Hilden, Germany). Quantitative real-time polymer chain reaction (qRT-PCR) was done with an SYBR Green kit (Thermo Fisher Scientific, Waltham, MA, USA) and Fluorescein Kit (BIOLINE, London, UK). The target genes were type I collagen (COL1A1), which represented a fibrogenic property, and type II collagen (COL2A1), which demonstrated a chondrogenic phenotype. The 18S rRNA was used as a housekeeping gene (Table 3).

## 3. Results

### 3.1. Viscosity of PLA Solutions

PLA solution viscosities were shown in Table 4. PLAL showed lower viscosity than PLAH at the same concentration. Increasing concentration led to a significant increase in viscosity. PLAmed solution at a lower concentration (6%) showed higher solution viscosity than PLAH (10%).

### 3.2. Fiber Morphology

#### 3.2.1. PLA Fiber Morphology

SEM micrographs of electrospun PLAL, PLAH, and PLAmed nanofibers, and their diameter distribution curves of electrospun fiber were shown in Figure 1 and Figure 2, respectively.

PLA fiber surfaces of each sample contained small pores randomly distributed on the fibers. PLAL−10 (Figure 1a–c) showed a large number of droplets or beaded particles instead of a fibrous structure. For PLAL−15 (Figure 1d,e), the beads still appeared on the electrospun fibers, or they can be called bead-on-string fibers, while PLAL−20 (Figure 1g–i) gave the porous fibers with no beads. PLAH−10 provided a small fiber diameter (Figure 2c) with high entanglement (Figure 1j–l). PLAH−15 exhibited the uniform fibers with random orientation (Figure 1m–o) when compared to other samples, while PLAH−20 (Figure 1p–r) gave larger diameter fibers (Figure 2e), but it was difficult to be processed. The higher molecular weight of PLA gave a larger fiber diameter of electrospun fibrous scaffold. PLAmed−6 (Figure 1s–u) provided the large fibers with the highest average diameter (Figure 2f) compared to electrospun PLAL and PLAH fibers.

#### 3.2.2. PLA/SF Fiber Morphology

Figure 3 shows the effect of SF content on PLA/SF electrospun fiber morphology. An electrospun PLA75:SF25 sample showed beads on the fiber surface (Figure 3a,b) with the average fiber diameter of 0.29 ± 0.20 µm (Figure 4a). A PLA50: SF50 sample (Figure 3d–f) showed a smoother fiber surface and smaller beads, with an average fiber diameter of 0.32 ± 0.26 µm (Figure 4b). For the PLA25:SF75 sample (Figure 3g–i), there was less fiber on the collector than in other samples. At this ratio, it tended to break up into electrospray instead, and it was difficult to process. The average fiber diameter was 2.19 ± 1.71 µm (Figure 4c). As SF content increased, the fiber diameter increased. With the addition of SF, the fiber diameter decreased compared to that of pure PLA fiber (Figure 1). PLAH50:SF50 was selected to compare with PLAmed50:SF50. For PLAmed50:SF50 (Figure 3j–l), the fine porous fiber and small beads appeared on the scaffold. The fiber diameter was 0.18 ± 0.08 µm (Figure 4d), which was smaller than that of PLAH50:SF50 fibers.

### 3.3. FTIR Spectra

FTIR spectroscopy in Figure 5 showed the strong characteristic absorption bands of SF powder that appeared at 1626 cm^−1^ (Amide I), 1512 cm^−1^ (Amide II), and 1228 cm^−1^ (Amide III). Electrospun pure PLA samples demonstrated significant absorption bands at 1752 cm^−1^ (the stretching vibration of carbonyl), 1452 cm^−1^ (the deformation vibration of a carbon hydrogen bond), 1368 cm^−1^, 1261 cm^−1^ (the antisymmetric stretching vibration of carbonyl), 1184 cm^−1^ (the stretching vibration of carbonyl), 1084 cm^−1^ (the antisymmetric stretching vibration of a carbon-oxygen bond), 868 cm^−1^, 755 cm^−1^, and 694 cm^−1^ (the bending vibration of carbon-hydrogen bond). Similar characteristic peaks for SF and PLA were observed in PLA/SF scaffolds (both PLAH and PLAmed), which confirmed the presence of both SF and PLA in the scaffolds.

### 3.4. Mechanical Properties

Tensile properties of PLAH−15, PLAH75:SF25, PLAH50:SF50, PLAmed−6, and PLAmed50:SF50 were shown in Table 5. PLAH−15 scaffold gave the highest tensile strength and highest Young’s modulus when compared to PLAH75:SF25 and PLAH50:SF50 scaffolds. After adding SF, the tensile strength of PLAH75:SF25 scaffolds decreased from 1.14 ± 0.09 MPa to 0.47 ± 0.07 MPa, while the elongation at break increased from 15.97 ± 1.80% to 26.22 ± 10.64% compared to that of pure PLAH−15. For PLAH50:SF50, the tensile strength was higher than PLAH75:SF25, but the elongation at break was lower.

The mechanical properties of PLAmed−6 scaffold showed higher values than the PLAmed50:SF50 scaffold. Incorporation of SF in PLA scaffolds slightly increased in elongation at break but decreased the tensile strength and Young’s modulus of electrospun scaffold.

### 3.5. Thermal Properties

The effect of SF content of PLA/SF scaffold on thermal degradation was shown in Figure 6. SF contained moisture content about 10% wt., while PLA showed very low moisture content. With the addition of SF, the moisture content in the sample slightly increased. The decomposition temperatures of PLA with the addition of SF shifted to a lower temperature due to the low thermal stability of SF.

Figure 7 shows DSC thermograms of electrospun scaffolds during the first heating scan (Figure 7a), cooling scan (Figure 7b), and second heating scan (Figure 7c). Thermal properties obtained from this figure were shown in Table 6 and Table 7. From the first heating scan which represented the thermal properties of the electrospun scaffolds obtained after the electrospinning process, the PLAmed−6 scaffold showed a higher glass transition, lower cold crystallization temperature, and higher melting temperature than the PLAH−15 scaffold. With the addition of SF into PLA, the shift to higher cold crystallization temperature and slightly higher melting temperature was observed in PLAH75:SF25 and PLAH50:SF50 scaffold. On the other hand, PLAmed50:SF50 showed lower cold crystallization temperature than pure PLAmed scaffold. However, after removing thermal history and cooling in DSC, no significant differences in glass transition temperature, cold crystallization temperature, and melting temperature among PLAH−15, PLAH75:SF25, and PLAH50:SF50 were observed from a second heating scan.

It is interesting to point out the difference in crystallization behavior between medical grade PLA and commercial grade PLA during a cooling scan. For PLAmed−6 and PLAmed50:SF50, a clear melt crystallization peak during cooling was presented. In contrast, no melt crystallization peak was shown for PLAH with and without SF. The %crystallinity of medical grade of PLA was higher than that of commercial grade of PLA both from the first heating scan and second heating scan. This indicated a higher ability to crystallize medical grade PLA than commercial grade PLA.

### 3.6. In Vitro Degradation

The effect of SF content on PLA/SF scaffold degradation was shown in Figure 8. Degradation profiles were shown as a percentage of residual weight of scaffolds by time. During the study period, both pure PLA and PLA/SF scaffolds degraded slowly. PLAH, PLAH75:SF25, and PLAmed exhibited very low degradation, while the weight of PLAH50:SF50 and PLAmed50:SF50 scaffolds decreased by time

### 3.7. Surface Wettability

The water contact angle of scaffolds measured from the droplet images was shown in Figure 9. The contact angle for electrospun pure PLAH and PLAmed scaffolds was highly hydrophobic with a contact angle of 142.49 ± 2.22° and 136.73 ± 2.09°, respectively. Electrospun PLAH:SF and PLAmed:SF scaffolds showed a slight decrease to 128.08 ± 1.10° (PLAH75:SF25), 112.81 ± 5.22° (PLAH50:SF50), and 127.45 ± 3.49° (PLAmed50:SF50). With the incorporation of SF, the PLA/SF scaffolds became more hydrophilic.

### 3.8. Cell Viability Test

MTT assay was carried out to evaluate HCPCs viability on PLA and PLA/SF electrospun scaffolds. Rather than the PLA/SF ratio and PLA grading, the effect of ethanol treatment (+) on cell viability is shown in Figure 10. Generally, the HCPC viability of ethanol treated scaffold groups (+) was higher than that of the non-treated groups. The cell viability in untreated scaffold groups was lower than the control group at every time point except PLA50:SF50, which presented higher viability compared to the control group at days 3 and 7. In the alcohol treated group (+), PLAH+ scaffolds had lower cell viability compared to medical grade PLA scaffolds. Cell viability at days 1 and 3 were slightly different among PLAH+ groups and, at day 7, PLAH−15+ had higher cell viability than PLAH50:SF50+ and PLAH75:SF25+, respectively. PLAmed−6+ had slightly higher cell viability than PLAmed50:SF50+ at every time point.

### 3.9. Quantitative Gene Expression

The gene expression analyses of HCPC seeded scaffolds were examined to demonstrate cellular phenotype. All groups of scaffolds were repeated with triplicate sample sets. The results were shown in Figure 11, and the expressions of COL1A1 of PLAmed−6 and PLAmed50:SF50 were higher than the cell without a scaffold at every time point. The expression of COL1A1 of PLAmed−6 was obviously higher than PLAmed50:SF50 (143.0 and 31.9) at day 7 and slightly higher at day 28 (41.2 and 38.1). On the other hand, PLAmed−6 and PLAmed50:SF50 had equivalent COL2A1 expression, which was higher than the control group at days 7 (1.4, 1.4) and 28 (2.0, 2.0). The highest expression of COL2A1 was observed at day 14 (2.0).

## 4. Discussion

For the electrospinning process, viscosity was an important solution parameter that affected the fiber formation ability [32]. Solution viscosity was affected by two important parameters, concentration [33] and molecular weight of polymer. An increase in concentration led to an increase in viscosity due to the longer chain entanglement of polymer. This can be seen from Table 4. PLAH showed higher viscosity than PLAL because of its higher molecular weight. The PLAmed−6 show higher viscosity when compared to PLAH−10. During polymer–solvent interaction, the solvent molecules go into the polymer and increase the chain mobility because of chain segmental relaxation [34]. The increase in the d-lactide content in the copolymer increases disorder in the polymer chains and reduces the crystallinity. Because of the less compact packing of d-lactide enantiomers, poly (d,l-lactide) is more amorphous, while poly (l-lactide) is enantiomerically pure polylactides [35,36]. The higher the l-lactide content in PLAmed (pure PLLA), the higher the solution viscosity [37]. The viscosity of PLAL at 10% and 15% concentration was not high enough to resist fiber deformation without defects at the given electric field and bead formation can be occurred. It could be assumed that the reduced viscosity resulted in an imbalance in viscous solution force and electrostatic force necessary for uniform fiber formation [38].

In electrospun fibers, the bead formation occurs when the surface tension in the charged jet is sufficient to change the jet into droplets to reduce surface area [39]. This is opposed by viscoelastic forces in the jet that resist changes to the fiber shape. In contrast, the increased viscosity of solution created higher viscoelastic forces that resisted the axial stretching during whipping, resulting in larger fiber diameter. This was confirmed by the increasing fiber diameter of PLA with increasing PLA concentration. However, for PLAH−20, the viscosity was too high so it was hardly spun and easily clogged the needle. Pure PLA scaffolds (both PLAH and PLAmed) displayed a wide fiber distribution in diameter when compared with those of the narrow distribution of PLA/SF scaffolds. PLA fibers contained small pores randomly distributed on the fibers owing to solvent evaporation [40]. Furthermore, the PLAmed−6 offers the biggest fiber diameter, which comes from complexity in the course of jet ejection [41].

With the incorporation of SF into the PLA, a gradient in solution viscosity and surface tension appeared during the flow of the emulsion solution. The PLA/SF fibers were drawn with more force, resulting in a comparatively thinner average diameter and distinct fibers [42]. The fiber diameter of PLAH/SF fibers with different ratios (PLAH75:SF25 and PLAH50:SF50) did not show any obvious difference. For the PLAmed50: SF50 sample, very thin fibers with beads and pores were produced along with nanofiber.

The FITR spectra with characteristic absorption peaks of SF powder, pure PLAH, and PLA/SF scaffolds were presented in Figure 5. Amide I (1626 cm^−1^) and amide II (1512 cm^−1^) formed the major bands in the SF protein structure that could be identified with the representative peaks (denoted by hashed lines), thus confirming the presence of SF on the surface of scaffolds [11,43,44].

The mechanical characteristics (Table 5) and structural integrity of the electrospun scaffold were important parameters for the meniscus tissue engineering application. The menisci serve many important biomechanical functions such as load transmission, shock absorption, stability, and joint congruity [45]. Mechanical strength is identified by the tension resistance of electrospun scaffolds in order to maintain their integrity of scaffold during implantation [46]. The mechanical properties of scaffolds were related to the morphology of electrospun fibers [47]. Apparently, the elongation at break of the PLAH/SF scaffold was improved by adding SF solution into the electrospinning solution. With the incorporation of SF content, the Young’s modulus and the tensile strength slightly decreased [31]. The mechanical properties of electrospun fibrous scaffolds were involved with their fiber density and junctions [48]. The mechanical properties of pure PLAmed scaffold showed the highest tensile strength and Young’s modulus, while the PLAmed/SF scaffold showed lower mechanical properties because of the beads and thin porous fiber. PLAmed (pure PLLA) of high molar mass has sufficient strength for use as load bearing material in medical applications [49].

Moreover, the lower d-Lactide content grades have a greater ability to crystallize [50]. d-lactide induces twists in the regular poly(l-lactide) molecular architecture. Molecular imperfections are responsible for the decrease in both the rate and extent of poly(l-lactide) crystallization [3]. The increase in %crystallinity is usually associated with the formation of bigger or more ordered crystals of the PLA, which can resist the higher tensile strength [51].

The tensile properties of scaffold augmented suture were reported as composed of elastic modulus (3.77 ± 2.81 to 16.90 ± 9.70 MPa), tensile strength (0.60 ± 0.43 to 3.40 ± 1.10 MPa), and %elongation (3.28 ± 1.49 to 20.09 ± 5.89) [52,53,54]. From our study, the Young’s modulus, %elongation, and tensile strength of both PLA and PLA/SF scaffolds were achieved. The %elongation at break of the scaffold obtained in our study can be used to confirm that the scaffold would not be cut through during repair [53]. Thus, our scaffold can be used to help tissue healing by seeding cells on scaffold incorporated suture. The cell seeded scaffold was shown to improve tissue healing and reduce gap formation at the repair site [55].

The effect of SF on thermal properties of PLA/SF scaffolds was shown in TGA thermogram in Figure 6. During the initial heating from room temperature to 250 °C, all samples had a mass loss due to the evaporation of water or solvent molecules. The water content increased significantly when SF was present. This can be ascribed to the better hydrophilicity and hygroscopicity of silk fibroin [56].

With the addition of silk fibroin, the decomposition temperature of the PLA/SF scaffolds tended to be lower. The decomposition rate gradually decreased with the increase of silk fibroin content [56]. From the DSC result, with an increase in SF, %crystallinity of the electrospun fibrous PLA/SF scaffolds decreased because the amorphous phase structure in the silk fibroin gradually increases [43,56,57]. Electrospun PLAmed scaffold had a high %crystallinity and a high melting temperature, which resulted in better mechanical properties than the PLAH scaffold. The cold crystallization peak upon cooling scan was observed in PLAmed and PLAmed/SF scaffolds, indicating that both specimens achieved a semi-crystalline state after the cooling [58]. This may be due to the higher l-lactide content in medical grade PLA [26]. PLAmed has the optical pure PLLA α crystalline form, which has a melting point range of approximately 170–180 °C [26,36]. Lower d-Lactide content meant a higher melting temperature [50]. The $T_{m}$ and degree of crystallinity are dependent on the purity of the PLA, the crystallization kinetics, and melting behavior of PLA [59].

Biodegradability is another important consideration of the scaffold. The scaffold must degrade in a desirable time to ensure proper tissue remodeling or regeneration [60]. The effect of SF content on PLA/SF scaffold degradation was shown in Figure 8. All scaffolds exhibited very low degradation. It can also be observed that the weight loss obtained is a function of the amount of silk fibroin present in the electrospun fibers [14]. PLA contains crystalline and amorphous regions. The long macromolecular chains segments are arranged more regularly and packed more strongly in the crystalline phase than in the amorphous phase. Thus, small molecules of water can attack the polymer chains in the amorphous phase more easily [61]. Adding SF can increase the amourphous region, which can increase the scaffold degradation. The degradation of silk fibroin was related to hydrophilic interaction as well as special nanostructures. In the degradation process, the hydrophilic blocks were firstly degraded [62]. The degradation of the PLA/SF scaffolds prepared in the present study was shown to be controllable by adjusting the PLA/SF ratio [7].

Surface wettability of biomaterials plays a critical role in influencing cell adhesion, cell proliferation, and cell migration [12]. To clarify the effect of SF and its content on the surface wettability of electrospun scaffolds, the water contact angle was measured. Water contact angle results (Figure 9) supported the fact that incorporation of SF into PLA scaffolds decreased the hydrophobicity of pure PLA scaffolds. The reason could be due to the presence of SF with naturally hydrophilic amino groups, carboxylic groups, and other functional groups in the backbone of SF [32,42].

Our results demonstrated that medical grade PLA electrospun scaffolds offered biodegradability and cellular induction for meniscus tissue regeneration. After seeding HCPCs which possess chondrogenic potential on the PLA and PLA/SF scaffolds, initial cell viability and the viability at days 3 and 7 were evaluated (Figure 10). Although higher SF content of PLA/SF scaffolds improved cell viability, the viability was still lower than the control group. Medical grade PLA scaffolds had higher cell viability, but ethanol treatment was mandatory to reduce toxicity of SF during its preparation [63]. Comparing between PLAmed−6 and PLAmed50:SF50, the cell viability was decreased in the PLA/SF composite scaffold. From gene expression analyses, COL1A1 which represented fibrogenicity was higher in PLAmed−6 at days 14 and 28, while COL2A1 was equal on day 7 and day 28 and higher in day 14. These presentations directed that the cells in PLAmed−6 scaffolds functioned toward more fibrogenicity, while the cells in PLAmed50:SF5 offered more chondrogenic properties. Since the HCPC seeded scaffolds were cultured in simple expanded media without any growth factor or mechanical stimulation, the cells could lose chondrogenic phenotype by time [64].

In the meantime, the properties such as smaller fiber diameter, more amino groups, and more hydrophilicity of electrospun PLA/SF fibrous scaffolds also have influenced the interaction between cells and scaffolds [14,32,42,57]. Treatment of scaffolds with 70% ethanol can increase the cell viability [65]. Alcohol treatment can reduce the toxicity of the residued electrospining solvents after fabricating because it effectively eliminates bacteria and viruses, and it can dissolve the chloroform and formic acid easily [66,67,68].

## 5. Conclusions

PLA/SF nanofibrous scaffolds were successfully fabricated by electrospinning. In this study, it was found that viscosity played an important role to determine the fiber formation ability, fiber morphology, and size. The average fiber diameter was increased along with an increase in solution viscosity. PLA concentration, structure, and molecular weight were three important parameters that affected the solution viscosity. PLAmed−6 scaffold gave a superior mechanical property, degradability, and cellular induction for meniscus tissue regeneration. However, for commercial non-medical grade PLA used in this study, it was not recommended to be used for medical application because of its toxicity. The addition of SF dominated the nanofiber morphology, diameter distribution, mechanical properties, decomposition temperature, %crystallinity, biodegradability, surface wettability, and cell viability. PLAmed50:SF50 scaffold has the potential to be used as biomimetic meniscus scaffold for scaffold augmented suture based on mechanical properties, cell viability, gene expression, surface wettability, and in vitro degradation.

## Figures and Tables

**Figure 1 polymers-14-02435-f001:**
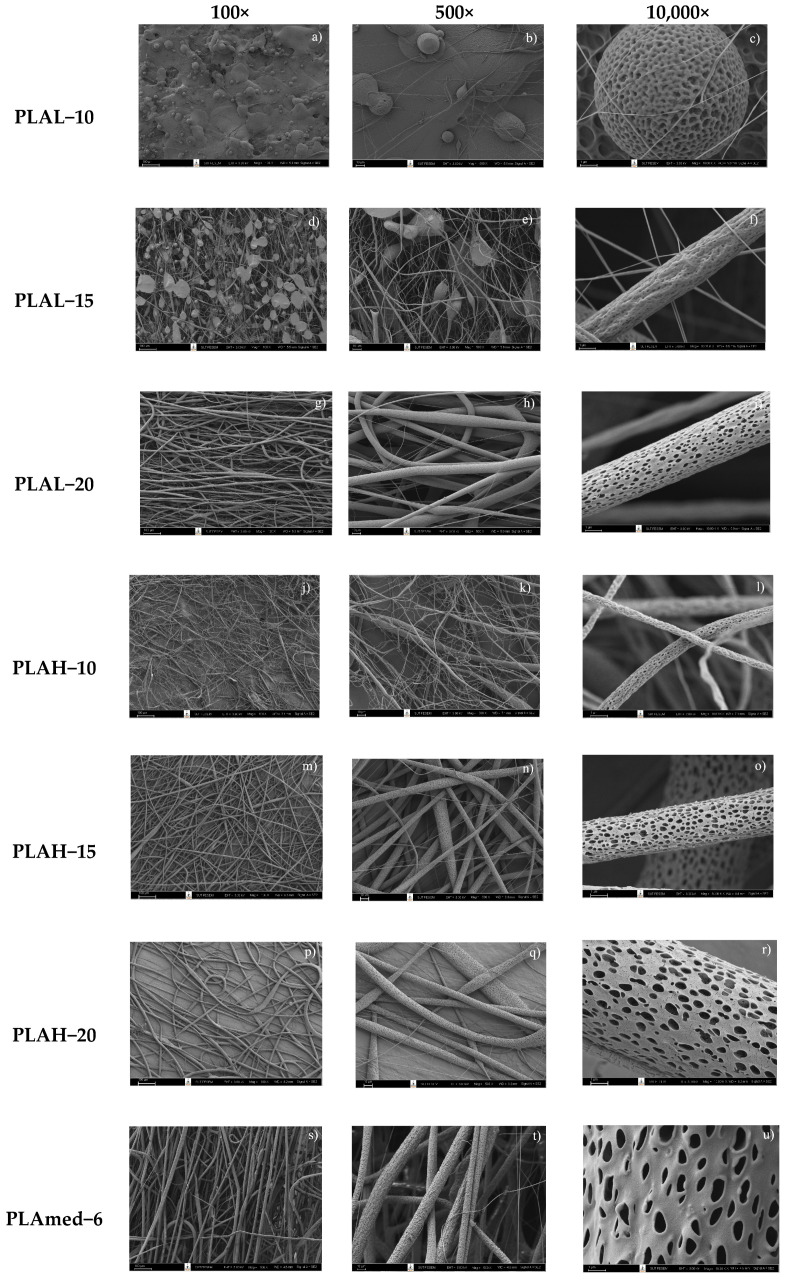
The morphology of pure PLAL, PLAH, and PLAmed fibers, which was fabricated by using various concentrations; (**a**–**c**) PLAL−10; (**d**–**f**) PLAL−15; (**g**–**i**) PLAL−20; (**j**–**l**) PLAH−10; (**m**–**o**) PLAH−15; (**p**–**r**) PLAH−20; and (**s**–**u**) PLAmed−6 at different magnifications (100×, 500×, and 10,000×).

**Figure 2 polymers-14-02435-f002:**
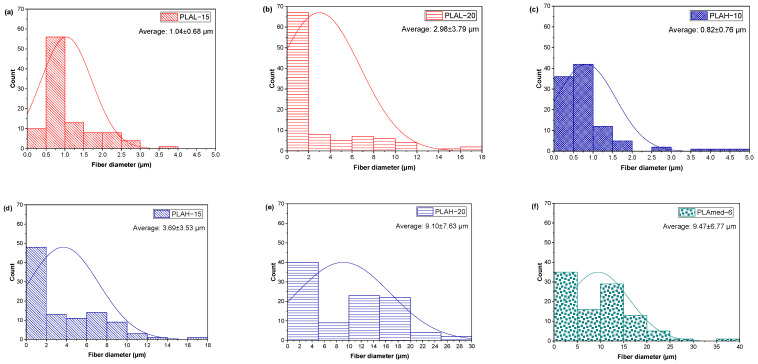
Histograms of electrospun fibers (**a**) PLAL−15; (**b**) PLAL−20; (**c**) PLAH−10; (**d**) PLAH−15; (**e**) PLAH−20; and (**f**) PLAmed−6.

**Figure 3 polymers-14-02435-f003:**
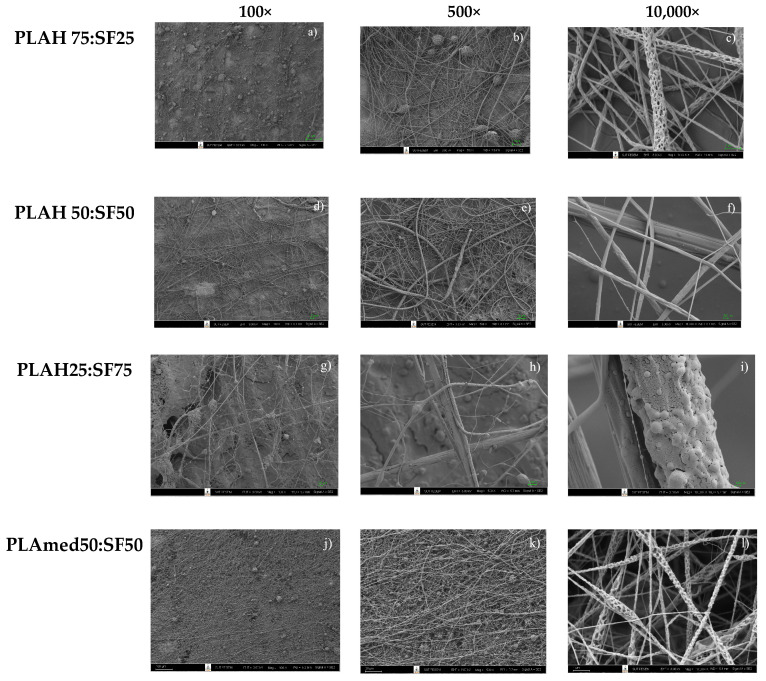
The morphology of (**a**–**c**) PLAH75:SF25; (**d**–**f**) PLAH50:SF50; (**g**–**i**) PLAH50:SF50; and (**j**–**l**) PLAmed50:SF50 fibers at different magnifications (100×, 500×, and 10,000×).

**Figure 4 polymers-14-02435-f004:**
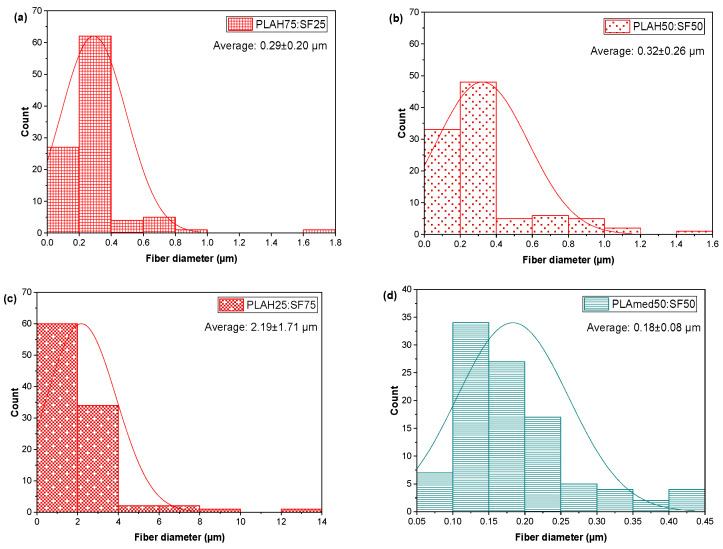
Histograms of electrospun fibers (**a**) PLAH75:SF25, (**b**)PLAH50:SF50; (**c**) PLAH25:SF75 and (**d**) PLAmed50:SF50.

**Figure 5 polymers-14-02435-f005:**
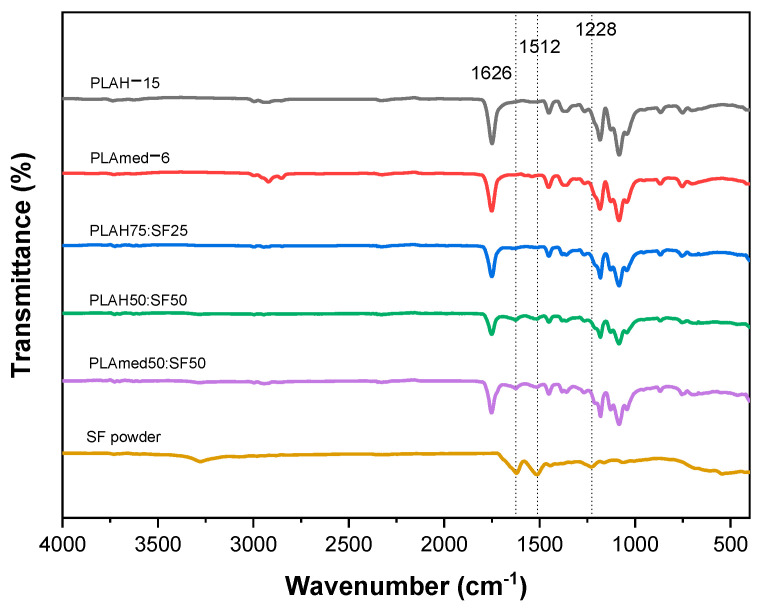
FTIR spectra of SF powder, PLAH −15, PLAH75:SF25, PLAH50:SF50, PLAmed−6, and PLAmed50:SF50 fibers.

**Figure 6 polymers-14-02435-f006:**
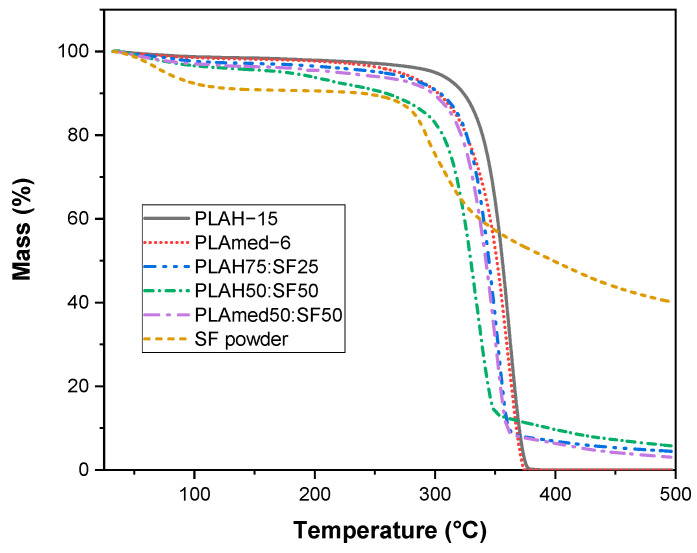
TGA thermograms of SF, PLAH−15, PLAH75:SF25, PLA50:SF50, PLAmed−6, and PLAmed50:SF50.

**Figure 7 polymers-14-02435-f007:**
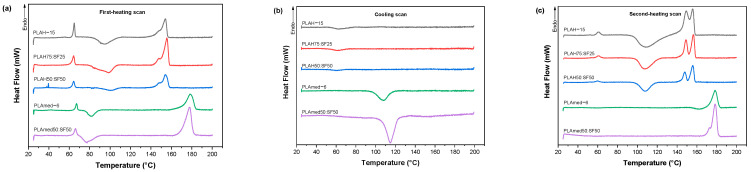
DSC thermograms from (**a**) first-heating scan, (**b**) cooling scan, and (**c**) second-heating scan of PLAH−15, PLAH75:SF25, PLAH50:SF50, PLAmed−6, and PLAmed50:SF50 scaffolds.

**Figure 8 polymers-14-02435-f008:**
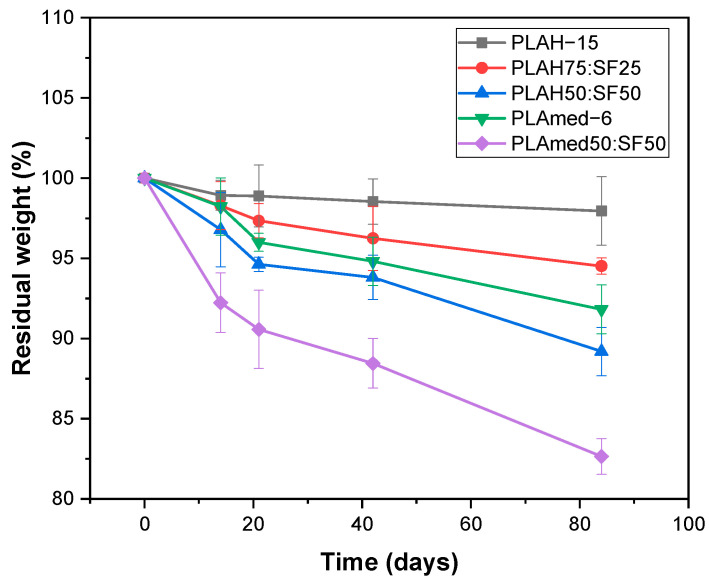
Residual weight (%) of electrospun scaffolds after degradation in phosphate buffered saline (PBS, pH = 7.4) as a function of time.

**Figure 9 polymers-14-02435-f009:**
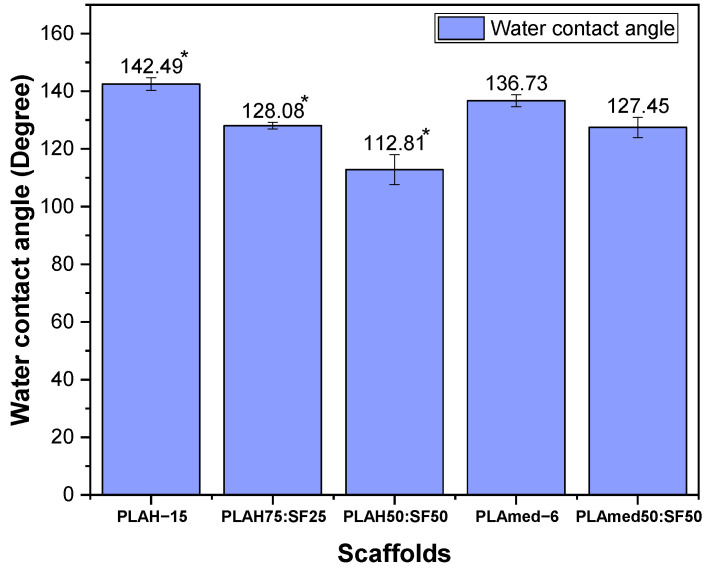
Water contact angle of PLAH−15, PLAH75:SF25, PLAH50:SF50, PLAmed−6, and PLAmed50:SF50 scaffolds; * From previous study [31].

**Figure 10 polymers-14-02435-f010:**
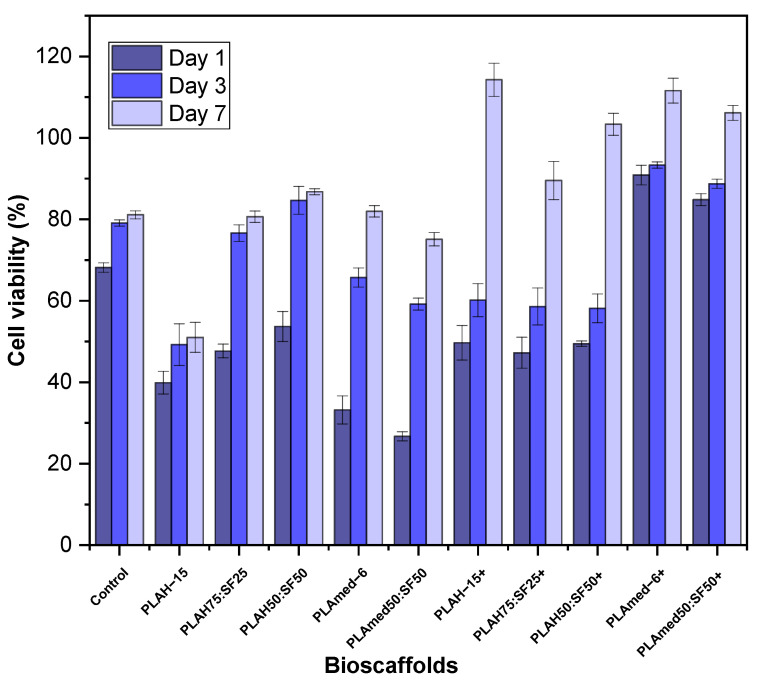
MTT assay for human chondrocyte cells viability on electrospun scaffolds after 1, 3, and 7 days. The plus sign (+) represents ethanol treated scaffolds (group 1).

**Figure 11 polymers-14-02435-f011:**
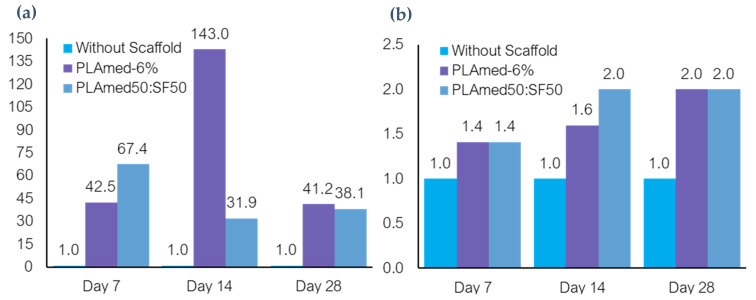
The expression of COL1A1 and COL2A1 at days 7, 14, and 28 were demonstrated in (**a**,**b**), respectively.

**Table 1 polymers-14-02435-t001:** PLA sample nomenclature.

Sample	Grade	Manufacturer	Molecular Weight (g/mol)	Concentration (% *w*/*v*)
PLAL−10	3251D	NatureWorks LLC	55,400 [21,23]	10
PLAL−15	3251D	NatureWorks LLC	55,400 [21,23]	15
PLAL−20	3251D	NatureWorks LLC	55,400 [21,23]	20
PLAH−10	4043D	NatureWorks LLC	127,300–147,400 [24,25]	10
PLAH−15	4043D	NatureWorks LLC	127,300–147,400 [24,25]	15
PLAH−20	4043D	NatureWorks LLC	127,300–147,400 [24,25]	20
PLAmed−6	L209S	Sigma-Aldrich LLC	177,000 [26,27]	6

**Table 2 polymers-14-02435-t002:** PLA/SF electrospun samples.

Samples	PLA Concentration (% *w*/*v*)	SF Concentration (% *w*/*v*)	PLA Ratio (Volume)	SF Ratio (Volume)
PLAH75:SF25	15	12	75	25
PLAH50:SF50	15	12	50	50
PLAH25:SF75	15	12	25	75
PLAmed50:SF50	15	12	50	50

**Table 3 polymers-14-02435-t003:** Sequences of the primer sets for qRT-PCR.

Genes		Primer Sequence (5′ to 3′)
Type I collagen	Sense	GGAGGAGAGTCAGGAAGG
(COL1A1)	Antisense	GCAACACAGTTACACAAGG
Type II collagen	Sense	GGCAGAGGTATAATGATAAG
(COL2A1)	Antisense	ATGTCGTCGCAGAGG
18S rRNA	Sense	ATACCGTCGTAGTTCC
	Antisense	GTCTCGTTCGTTATCG

**Table 4 polymers-14-02435-t004:** PLA solution viscosity for electrospinning.

PLA Solution (*w*/*v*)	Viscosity (cP)
PLAL−10	58.02 ± 0.80
PLAL−15	278.68 ± 5.47
PLAL−20	690.54 ± 30.18
PLAH−10	652.98 ± 36.15
PLAH−15	2304.73 ± 74.49
PLAH−20	9319.67 ± 91.40
PLAmed−6	1024.81 ± 83.15

**Table 5 polymers-14-02435-t005:** Mechanical properties of PLA based scaffolds.

Samples	Ultimate Tensile Strength (MPa)	Elongation at Break (%)	Young’s Modulus (MPa)
PLAH−15	1.14 ± 0.09 *	15.97 ± 1.80 *	70.21 ± 4.99 *
PLAH75:SF25	0.47 ± 0.07 *	26.22 ± 10.64 *	20.51 ± 3.97 *
PLAH50:SF50	1.05 ± 0.43 *	17.51 ± 3.89 *	16.49 ± 8.71 *
PLAmed−6	2.06 ± 0.28	14.34 ± 1.00	109.38 ± 12.21
PLAmed50:SF50	0.85 ± 0.11	12.14 ± 2.20	22.59 ± 6.59

* From previous study [31].

**Table 6 polymers-14-02435-t006:** Thermal properties of PLAH−15, PLAH75:SF25, PLAH50:SF50, PLAmed−6, and PLAmed50:SF50 obtained from DSC thermogram during first-heating and cooling scan.

Samples	First-Heating Scan							Cooling Scan		
	$T_{g\;}\left({}^{\circ}C\right)$	$T_{cc\;}\left({}^{\circ}C\right)$	$\Delta H_{cc}\;$ $\left(Jg^{-1}\right)$	$T_{m_{1}}\;\left({}^{\circ}C\right)$	$T_{m_{2}\;}\left({}^{\circ}C\right)$	$\Delta H_{m}\;$ $\left(Jg^{-1}\right)$	$\chi_{c\;}\left(\%\right)$	$T_{g}\;\left({}^{\circ}C\right)$	$T_{c}\;\left({}^{\circ}C\right)$	$\Delta H_{c\;}$ $\left(Jg^{-1}\right)$
PLAH−15	64.90	94.22	14.81	148.05	154.07	20.53	21.94	63.68	-	-
PLAH75:SF25	64.38	98.44	13.35	147.81	155.49	25.86	21.83	62.66	-	-
PLAH50:SF50	64.32	100.55	9.09	148.12	154.19	21.41	12.81	61.55	-	-
PLAmed−6	67.19	82.01	12.00	-	178.61	42.61	45.52	-	107.91	19.01
PLAmed50:SF50	66.03	77.42	8.56	-	177.82	42.71	25.55	-	114.84	30.52

**Table 7 polymers-14-02435-t007:** Thermal properties of PLAH−15, PLAH75:SF25, PLAH50:SF50, PLAmed−6, and PLAmed50:SF50 obtained from DSC thermogram during second-heating scan.

Samples	Second-Heating Scan						
	T_g_ (°C)	T_cc_ (°C)	$\Delta H_{cc}\;$ $\left(Jg^{-1}\right)$	$T_{m_{1}}\;\left({}^{\circ}C\right)$	$T_{m_{2}}\;\left({}^{\circ}C\right)$	$\Delta H_{m}\;$ $\left(Jg^{-1}\right)$	$\chi_{c\;}\left(\%\right)$
PLAH−15%	59.28	108.78	14.48	149.42	155.76	21.53	23.00
PLAH75:SF25	58.55	107.67	24.21	149.03	156.40	25.71	21.70
PLAH50:SF50	57.41	107.79	21.58	147.85	155.92	23.71	14.19
PLAmed−6%	67.57	-	-	-	178.33	39.97	42.71
PLAmed50:SF50	63.94	-	-	172.50	178.41	41.04	24.56

## Data Availability

Not applicable.

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
