# Peer review of "Electrospun Poly(lactic acid) and Silk Fibroin Based Nanofibrous Scaffold for Meniscus Tissue Engineering"

_polymers, 2022, doi:10.3390/polym14122435_

Round 1

Reviewer 1 Report

The manuscript reported PLA/SF composite biomimetic meniscus scaffold developed by electrospinning technique. The concept is not new, and some results are questionable

It is well known that the PLA used as bio-material for scaffold is not the commercial products used for packaging from Naturework, It should have higher molecular weight with special catalyst that is not harmful for cells. The mechanical properties listed on Table-4 have clearly indicated that such poor mechanical properties are far from the requirement from scaffold application.

Author Response

Eventhough commercial PLA as bio-material is usually used  for packaging not for scaffold. There is no published research paper that showed the toxicity or cell viabiliy results with comparable molecular weight.  Generally, molecuar weight plays a significant role in mechanical properties. So our finding result of the poor mechanical properties of the commercial PLA compared with medical grade PLA with comparable molecular weight is worth to be noted. Also the difference in D and L lactide content seems to be more influences than molecuar weight effect in this case and this is mentioned in the paper.

Reviewer 2 Report

I got this manuscript for review for the second time. It has been corrected by the authors and resubmitted. The manuscript has not been corrected properly in total and still have many points to be modified with respect to both my comments and the other reviewer's. However, some points of the revised manuscript are getting better.

Referring to my comments from the previous round: the Authors added the information about structure of PLA used, but the explanation of the differences in the properties with respect to the different structures is not always complete and sufficient.

I stand by my opinion that the article is worth publishing. 

Author Response

Thank you very much for your support. We tried to correct according to your suggestion as shown in the attached file. The highlight sentences are the additional information and discussion suggested by the reviewers especially for the point of  D and L lactide monomer that affects the physical properties.

Reviewer 3 Report

This paper studied a pretty old concept of making composite using commercial poly(lactic acid)(PLA) and silk fibroin (SF) by electrospun and applied in tissue engineering. In literature, several papers already exist on SF-PLA composites. This work is not bringing anything new to them. There is nothing novel in this work and it is weakened by publication from their own group. However, the author executes the studies nicely and the authors should get an appreciation for that. I would suggest publishing this paper in a more appropriate journal such as “Materials” instead of “Polymers”. Since this work is not adding or exploring something which is solely focused on polymer. 

Author Response

Thank you very much for your suggestion. We would be grateful if this can be published in Polymers because  we thought that our findings would benefit the readers who are interested in the polymer applications in the  special Issue of Functionalization and Medical Application of Polymer Materials.

Round 2

Reviewer 1 Report

I think the revised manuscript has significantly improved and should be considered as accept.

Reviewer 3 Report

The manuscript can be accepted in its present form.

This manuscript is a resubmission of an earlier submission. The following is a list of the peer review reports and author responses from that submission.

Round 1

Reviewer 1 Report

This paper describes the use of electrospun poly(lactic acid)(PLA) and silk fibroin (SF) composite fibers for tissue engineering. By changing the composition of PLA and SF, the author looked at the change in different physical properties of the composites. In terms of applications, they looked at cell viability and gene expression in vitro. However, the novelty of this study was weakened by various previous publications from other groups and publications from their own group (Effect of Silk Fibroin Content on Physical and Mechanical Properties of Electrospun Poly(lactic acid)/Silk Fibroin Nanofibers for Meniscus Tissue Engineering Scaffold doi:10.1088/1742-6596/2175/1/012016). The more serious thing is that some of the data are exactly copied from this previous publication without reproducing for the current manuscript -

Figure 3 images (d-f) PLAH50:SF50 of this paper are copied from the published manuscript (Figure 3 g, h, i)

Table 4 some of the values of samples is exactly the same as in the previous paper (Table 2) with exact error value.

Figure 9 some of the contact angle values are exactly the same as in the previous manuscript (page 6) with exact fractions.

Now, let's back to science. As I said before, there are several papers already exist on SF-PLA composites. The author has not talked about them in the introduction. Either they don’t know the current state of the literature on SF-PLA composites or they have intentionally avoided them. Whereas, they have spent a detailed paragraph on “meniscus” which is not required in the introduction. The introduction should be more on what has been done with this material and what are the new things current work bringing in.

The PLAH and PLAMed polymer have similar molecular weights whereas, their viscosity is significantly different. Can the author explain it?

Although the application part of the manuscript is done nicely, the paper is not up to the level of novelty and significance of the Polymers. 

Reviewer 2 Report

In general, the results are interesting, proper introduction and experimental design are provided. However, data interpretation should be improved and some raw data should be added as supporting information. After a few inaccuracies and a minor revision were made, the article could be published.

Detailed comments:

  1. Comparing the solution viscosity of PLA, thermal properties, as well as other properties, it would be necessary to refer to the structure of the PLA used and explain the differences in this aspect (Resomer L2009S is a L-PLA, while the other two polyesters contain units of the L and D configurations)
  2. Figures 2 and 4. How was the diameters determined and the histograms taken? How many fibers were measured and with what conditions? Editing errors in Figures 4a and 4c should be eliminated.
  3. Figure 8. Statistics are missing in the degradation studies. How many samples were tested and what was the error of determination. This is especially important due to the slight differences in the degradation rate

Reviewer 3 Report

The manuscript reported PLA/SF scaffold. Both concept and system are not new. There is no valuable discussions. Specific comments are:

  • Why compared commercial and medical grade PLA since it is well known that the toxicity of the commercial PLA from catalyst.
  • As a scaffold application, what is the solvent chloroform remained in the materials, even PPM?
  • One of the key weaknesses of application of PLA-based materials as scaffold is still the poor mechanical properties. It is seen that tensile properties were decreased significantly after additional of SF (see Table-4).